# Body Weight Changes During Childhood and Predictors of Excessive Body Weight in Adolescence—A Longitudinal Analysis

**DOI:** 10.3390/nu16244397

**Published:** 2024-12-21

**Authors:** Aleksandra Lemanowicz-Kustra, Michał Brzeziński, Magdalena Dettlaff-Dunowska, Anna Borkowska, Maciej Materek, Kacper Jagiełło, Agnieszka Szlagatys-Sidorkiewicz

**Affiliations:** 1Department of Pediatrics, Gastroenterology, Allergology and Nutrition, Faculty of Medicine, Medical University of Gdańsk, 80-803 Gdańsk, Poland; brzezinski@gumed.edu.pl (M.B.); magda.dettlaff@gmail.com (M.D.-D.); andzia@gumed.edu.pl (A.B.); agnieszka.szlagatys-sidorkiewicz@gumed.edu.pl (A.S.-S.); 2Department of Pediatrics, Copernicus Hospital, 80-803 Gdańsk, Poland; maciej.materek@gmail.com; 3Department of Preventive Medicine and Education, Medical University of Gdańsk, 80-211 Gdańsk, Poland; kacper.jagiello@gumed.edu.pl

**Keywords:** excess body weight, overweight, obesity, children, Poland, predictors, KPRT

## Abstract

**Introduction:** Abnormal body weight, including overweight and obesity, is a common health problem affecting children and adolescents. The present study aimed to analyse weight changes in children from preschool age to adolescence and to identify early predictors of excessive weight in adolescence, such as blood pressure and physical fitness observed in preschool children. **Methodology:** Data from 3075 children (1524 girls and 1594 boys), collected as part of the Gdańsk Centre for Health Promotion’s “Your Child’s Healthy Life” programme, were analysed, with each child assessed at ages 6, 10, and 14. **Results:** The results indicated that boys were more likely to be overweight, with a tendency for obesity to increase with age. Children who were overweight or obese at age 6 had a higher risk of remaining so for a longer period of time. In addition, low physical fitness (as measured by the KPRT test) and elevated blood pressure were significantly associated with excess body weight. **Conclusions:** The study underscores the importance of early intervention and consistent monitoring of childhood overweight and obesity to reduce their long-term impact on health.

## 1. Introduction

Abnormal body weight is a global health problem affecting different age groups, including children and adults. According to a WHO report, there has been a significant increase in the proportion of overweight and obese children worldwide. The prevalence of excess body weight (EBW) among children and adolescents aged 5–19 years has increased from as low as 4% in 1975 to just over 18% in 2016. This increase was similar for both boys and girls [1]. According to the PITNUTS survey report (2016), approximately 10% of 1- to 3-year-olds in Poland were overweight/obese and 18.4% were at risk of being overweight [2]. In contrast, among 8-year-olds, the problem of overweight and obesity affected almost one in three children [3]. The most recent results of the study of health behaviours in adolescents aged 11–15 years (international HBSC study) indicated that 29.7% of boys and 14.3% of girls were overweight, based on 2007 WHO criteria. It should be noted that these percentages are several percent higher compared to the results of the 2014 survey [4].

The childhood obesity epidemic has become one of the major challenges for modern society due to its numerous consequences for health. The most common complications of abnormal body weight in children can be classified into two categories: childhood complications, including psychiatric, metabolic, cardiovascular, and respiratory disorders, and adult complications, including premature mortality, reduced fertility, and socioeconomic complications [5,6,7,8]. Due to the numerous complications associated with excessive body weight, early implementation of comprehensive long-term specialised care aimed at sustainable behavioural changes is crucial [9]. Early education focused on risk factors for abnormal body weight and modification of environmental factors leading to its development, such as diet and physical activity, is an important approach [10,11]. Multiple studies confirmed that a reduction in physical activity is associated with an increase in the prevalence of abnormal body weight [12,13]. It should also be taken into account that a reduction in physical activity may also occur as a secondary result due to mobility difficulties among overweight or obese individuals, which in turn leads to a further increase in body weight. Preventive examinations, such as the regular, routine child health check-ups in Poland, which are used to assess, among other things, body weight, height, and blood pressure, are helpful in detecting excessive body weight [10], but they are not used to assess physical activity.

Therefore, the aim of this study was the longitudinal analysis of weight changes in children between preschool age and adolescence, and the identification of potential predictors of excessive body weight in adolescence, such as body composition, blood pressure, and physical fitness, which can be found in preschool children.

## 2. Materials and Methods

The data of a total of 3075 children (1524 girls and 1594 boys), collected as part of the screening study conducted in these age groups by the Gdańsk Centre for Health Promotion “Your Child’s Healthy Life”, were analysed. The main aim of the programme is to assess the health and healthy lifestyle, physical development, and identify the risks to children’s health (programme financed by the City of Gdańsk).

Each child was examined 3 times at the ages of 6, 10, and 14 years. At each of the 3 visits, a history of chronic diseases was taken, and anthropometric measurements, blood pressure, and physical capacity were assessed based on the Kasch Pulse Recovery Test (KPRT). Children with comorbidities or absolute contraindications to exercise due to cardiovascular overload were excluded from the study on the basis of a parent or guardian interview, physical examination, and data from medical records.

### 2.1. Measurements

Body height was measured to the nearest 1 mm (children standing barefoot in the Frankfurt position), and body weight to the nearest 100 g (children were barefoot, wearing only underwear or gym clothes). Based on the collected measurements, the BMI was calculated according to the formula: BMI = body weight in kg/height^2^ m^2^. The BMI values were compared to the OLAF national centile grids [14]:

BMI ≤ 5 pc underweight

BMI 5–85 pc normal body weight

BMI ≥ 85 pc < 95 pc overweight

BMI ≥ 95 pc obesity

Due to the lack of centile values using the above grids for the age range between 6.00 and 6.49 (OLAF centile grids are for ages 6.5–18.5), it was necessary to assign centile values for children aged 6.00–6.49 using centile values for an age equal to 6.50. The term used in this study is “excessive body weight”, which is assumed to include both overweight and obese children (Mensor WE150 (Mensor AJ, Warsaw, MAZ, Poland)).

### 2.2. Blood Pressure Measurement

Blood pressure was measured using an Omron device (OMRON M3, Omron Healthcare, HongKong, China) placed on the left arm, with an appropriately sized cuff, following a 5 min rest. Measurements were taken 3 times, with the mean of the measurements subsequently recorded. Hypertension was diagnosed with values ≥ 95 pc based on the OLAF national centile grids [15].

### 2.3. Assessment of Cardiovascular Fitness

The 3 min step test, the Kasch Pulse Recovery Test, was used to assess physical fitness. The test consists of rhythmically ascending and descending a 30 cm step for 3 min at a rate of 24 ascents and descents per minute and assessing the average post-exercise heart rate. Heart rate (HR) was monitored continuously using an electronic analyser “Polar” (Polar T31, Polar Electro, Kempele, Finland) for 3 min of exercise and for 1 min and 5 s of rest in a sitting position. Only the post-exercise heart rate recorded during 1 min, starting 5 s after the end of the test, was analysed. Based on the arithmetic means calculated from these values, physical fitness was assessed based on a reference system using a 6-point scale with a division into good fitness ≤ 3 (1, 2, 3) and poor fitness ≥ 4 (4, 5, 6). For each participant, the decision to qualify for the exercise test was made by the physician. The test was stopped if the exercise heart rate exceeded 180 beats per minute for more than 15 s.

All procedures of this study were approved by the Bioethics Committee for Scientific Research of the Medical University of Gdańsk.

### 2.4. Statistical Analysis

Results in the tables are presented as counts (N) and percentages (%). Standard deviation score (SDS) trajectories were used to present changes in quantitative data for the BMI category at successive time points. Visualisation of the percentage data was performed using bar charts. BMI categorisation was based on OLAF centile grids for the Polish population. A chi-square test of independence was used to examine the association between genders. The level of significance in this study was taken as alpha = 0.05. The analysis was performed using the software package R version 3.6.3.

## 3. Results

A total of 3075 children, 1510 girls (49.1%) and 1565 boys (50.9%), were included in the study. BMI analysis of the study participants showed 336 children were overweight at 6 years of age (49.40% M and 50.6% F) and 189 were obese (51.85% M and 48.15% F). At 10 years of age, 427 children were overweight (54.56% M and 45.43% F) and 195 were obese (50.89% M and 49.11% F), and at 14 years of age, 505 children were overweight (55.64% M and 44.36% F) and 211 were obese (56.87% M and 43.13% F; Table 1). There was no statistically significant difference between genders in the prevalence of overweight and obesity in subsequent age groups (Table 1).

On the basis of the collected data, gender-specific trajectories of BMI SDSs were determined, separately for girls and boys, in the subsequent years of follow-up. The analysis showed that the mean BMI SDS (BMI SDS defined as obesity (+2)) in the group of obese children aged 6 years in the later years decreased in both boys and girls (Figure 1 and Figure 2).

Children with normal weight at 6 years old remained within their centile channel in subsequent years of observation. In contrast, in underweight children, the mean SDS decreased over time, meaning that their BMI gradually increased and approached the population mean value. As a result, the deviation from the mean gradually decreased, indicating a levelling off of the differences from their peers.

The BMI of children at 10 and 14 years of age was analysed in relation to the BMI groups the children had been categorised into at 6 years of age. The results are shown in Table 2 (girls) and Table 3 (boys).

The girls who were underweight at 6 years of age remained in that category or had a normal body weight at the ages of 10 and 14 (100%). In this group, there was no overweight or obesity at the age of 14. In the group of boys who were underweight at the age of 6, a total of 5.3% had excess body weight (EBW) at 14 years old. In the group of normal-weight girls at 6 years of age, there was a significant increase in the percentage of EBW (BMI > 85 pc) to 9.2% at the age of 10 and 13.3% at the age of 14. In the group of normal-weight boys, EBW was reported in 10% at the age of 10 and in 16.1% at the age of 14. Only one of the girls who were overweight at the age of 6 was in the lowest BMI category at the age of 10. None of those girls were classified as underweight at the age of 14. However, a normalisation of body weight in this group was observed in 40.0% at the age of 10 and in 44.1% at the age of 14. In 18.8% (32 subjects) at the age of 10, an increase in body weight to obesity was observed, whereas the same was observed in 21.2% (36 subjects) at the age of 14. In the group of boys who were overweight at the age of 6, normalisation of body weight occurred in 30.7% at the age of 10 and in 33.1% at the age of 14. Obesity, on the other hand, was observed in 21.1% (35 subjects) at the age of 10 and in 24.1% (40 subjects) at the age of 14. Girls who were obese as 6-year-olds still overwhelmingly had an excessive body weight at 10 years old, and this concerned 89% of the subjects in this group. A decrease in body weight was not observed until the age of 14, when abnormal body weight was found in 68.2% of the subjects. Obese boys at the age of 6, similarly to girls, had EBW at the age of 10 (94.9% of the subjects), while in 91.8% of the subjects, no significant decrease in body weight was observed at the age of 14. The above changes in body weight for the BMI groups into which the subjects were categorised at 6 years old are shown in Figure 3 and Figure 4.

We analysed whether abnormal KPRT (Kash Pulse Recovery Test) results at age 6 years correlated with abnormal body weight among older children. According to the collected data, 56.9% of overweight children and 47.8% of obese children had normal blood pressure at 6 years of age, while 62.9% of overweight children and 81.1% of obese children at 6 years of age had abnormal results in the KPRT (Table 4). We found a statistically significant relationship between body weight and the KPRT result in all age groups studied, with abnormal body weight (overweight or obesity) being associated with worse KPRT results (*p*-value for each age group < 0.001; Table 4). Blood pressure measurements revealed that the higher the body weight was, the worse the blood pressure (*p*-value for each age group < 0.001).

A poor result in the cardiovascular fitness test at the age of 6 years old was mostly associated with excessive body weight at follow-up in both genders (Figure 5 and Figure 6).

We also observed that overweight and obesity at the age of 6 years correlated with abnormal KPRT results in later years (at 14 y.o.). Children who were overweight and obese at 6 years old were more likely to have abnormal KPRT results at 14 years old compared to normal-weight children (Figure 7 and Figure 8).

Obese children who had a good KPRT result at the age of 6 were less likely to remain obese at the ages of 10 and 14 compared to children who had poor KPRT results (Table 5).

## 4. Discussion

Nutritional disorders are currently one of the key health, social, and psychological problems worldwide [1,16,17]. The aetiology of obesity is complex—both genetic predisposition and environmental factors, such as diet and physical activity, are major contributors. The number of overweight and obese children is steadily increasing, which may lead to an increase in complications related to abnormal body weight among children.

In Poland, there are discrepancies in the data on the increasing trends of overweight and obesity among children. Brzezinski et al. reported a slight increase among children aged 6–7 years born between 1987 and 1999 but showed a periodic increase in the prevalence of overweight among children aged 6–7 and 12 years born between 1993 and 1997 [18]. Chrzanowska et al. documented a significant increase in mean body weight based on data from children studied in 1971 and 1983 [19]. Another Polish study conducted in Cracow revealed a more pronounced increase in the mean body weight and height of subjects studied in the 1970s and 1980s than in subsequent decades, with overweight and obesity more common in boys than girls [20]. In our study population, we observed an increasing trend in EBW: at the age of 6 years, 8.6% of the boys and 8.5% of the girls had EBW, at the age of 10 years, 10.8% of the boys and 9.5% of the girls, and at the age of 14 years, 13% of the boys and 10.2% of the girls. Overweight and obesity were more common among boys, especially at school age (Table 1). Similar conclusions were reached by Czech researchers, who found that overweight and obesity were more common in boys aged 6–7 years (7.9% and 8.7%) than in girls (7.7% and 7.5%, respectively) [21].

Our observations are consistent with the results of a meta-analysis by Simmonds et al., who, based on 15 prospective cohort studies from around the world, showed that obese children and adolescents were about five times more likely to be obese in adulthood than children who were not obese. Approximately 55% of obese children went on to be obese in adolescence, 80% of obese adolescents remained obese in adulthood, and 70% remained obese after the age of 30 [22]. A similar pattern was described by Reilly et al., who found that 34% of children overweight at the age of 7 became obese by the age of 13 [23]. It is noteworthy that obesity is a contributory factor in the development of metabolic syndrome, including cardiovascular disease, hypertension, and diabetes [24,25].

A number of studies indicated that in adulthood, there is a higher incidence of cardiovascular disease and type 2 diabetes in individuals who had a BMI above the 75th percentile in childhood [26,27,28], and the prevalence of high BP among children worldwide increases in parallel with increasing body weight [27,29,30]. This is supported by the results of a study by E. Szczudlik et al., who showed that a higher risk of cardiovascular disease was associated with a younger age (<5 y.o.) of obesity diagnosis. In addition, this study showed that almost 90% of obese participants were diagnosed with high BP [31].

According to W. Kowalewski et al., 11-year-old children with excessive body weight have systolic hypertension 1.14 times more often and diastolic hypertension 1.3 times more often than their peers with normal body weight [32]. In our study, we confirmed that EBW is a major health problem leading to hypertension. Hypertension occurred in 46.4% of the 6-year-olds, in 61.1% of the 10-year-olds, and in 45% of the 14-year-olds with EBW. Therefore, body weight should be systematically monitored among children, since, according to our data, 13.3% of girls and 16.2% of boys with normal body weight at age 6 had EBW at age 14 (Table 2 and Table 3). In a study carried out by Jankowska et al. on a group of 591 obese children, as many as 12.9% of children aged 9–12 years were diagnosed with metabolic syndrome [24].

During our observations, we found that children with EBW demonstrated poor physical fitness. The study population underwent a three-minute pulse recovery test (KPRT). We found a statistical correlation between body weight and the KPRT result in all age groups—excessive body weight was associated with poorer KPRT results (*p*-value for each age group < 0.001). Similar observations were made by Gupta et al., who found poor KPRT results in 14% of the Indian children aged approximately 12 years who were overweight and obese [33]. A study by Moczulska et al. indicated that similarly to children, overweight and obese adults also demonstrated higher post-exercise heart rates as well as a greater degree of post-exercise dyspnoea [34].

In our study, we found that poor physical fitness test results at the age of 6 were associated with excessive body weight at a later age. Children with good KPRT results were less likely to be obese at 10 and 14 years of age compared to children with poor KPRT results, indicating the importance of using fitness tests from an early age (Table 5). M. Dettlaff-Dunowska et al. demonstrated that better KPRT results were correlated with a loss of adipose tissue in an intervention programme for children [35].

Logistic regression models (enclosed with Appendix A) it appears that a poor KPRT score significantly increased the risk of overweight and obesity in children: among 6-year-olds by more than 3 times (odds ratio OR = 3.2) and 10- and 14-year-olds by more than 2.5 times (OR = 2.58 for 10 y.o. and OR = 2.65 for 14 y.o.), compared to children with a good KPRT score. The results indicated that low physical fitness is a key risk factor for overweight and obesity. It is worth paying attention to preventive measures that promote physical activity from an early age to improve KPRT scores and reduce the risk of weight problems. In addition, in the study group, being a boy was associated with a higher risk of excessive body weight compared to girls: among 6-year-olds by 26%, 10 y.o. by 17%, and 14 y.o. by 19%. This may suggest the need to differentiate interventions by gender.

Our study benefited from the multiannual follow-up of the study cohort, which allowed for a longitudinal analysis of changes in BMI SDSs during childhood. We found that the children who were obese at the age of 6 years lost weight over time, while underweight children gained weight, and normal-weight children maintained it (trajectory table). In the present study, we have shown that between 6 and 14 years of age, there was little change in nutritional status in the below-normal BMI children, which is a desirable result. Unfortunately, a similar pattern occurred in overweight children: the majority of children who were obese at the age of 6 still had an abnormal body weight at the age of 10 (89% of girls and 94.9% of boys; Table 2 and Table 3).

## 5. Conclusions

The long-term follow-up of the study cohort revealed that excessive body weight in early childhood tends to persist into adolescence, emphasising the need for early intervention and health education. We found that obese children who were physically fit at age 6 were less likely to remain obese by age 10 and in adolescence. This suggests that KPRT assessment at age 6 could be a useful tool for identifying children in need of intervention programs. Early identification of at-risk children allows for timely and targeted strategies to prevent the long-term consequences of obesity.

These findings underline the importance of monitoring children’s physical development from an early age to implement timely interventions and prevent obesity. Early intervention and health support are crucial in managing overweight and obesity in children. Promoting physical activity and healthy eating habits at a young age can help establish long-term healthy behaviours. Future research should focus on evaluating the effectiveness of early intervention programs to determine the most impactful strategies for combating childhood obesity.

## 6. Limitation of the Study

The study was conducted in a single city, Gdańsk, which may limit the generalisability of the results to children and adolescents from other regions or countries.

Although the study followed the children for several years (at ages 6, 10, and 14), a longer follow-up period might provide more insight into the long-term effects of early obesity and the effectiveness of interventions.

The study focused on physical fitness and blood pressure as predictors of overweight, but did not take into account other factors, such as socioeconomic status, diet, or genetic predisposition, which might provide a more complete understanding of the causes of childhood obesity.

## Figures and Tables

**Figure 1 nutrients-16-04397-f001:**
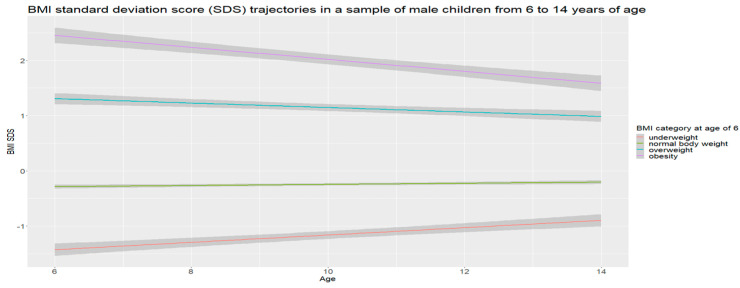
Boys.

**Figure 2 nutrients-16-04397-f002:**
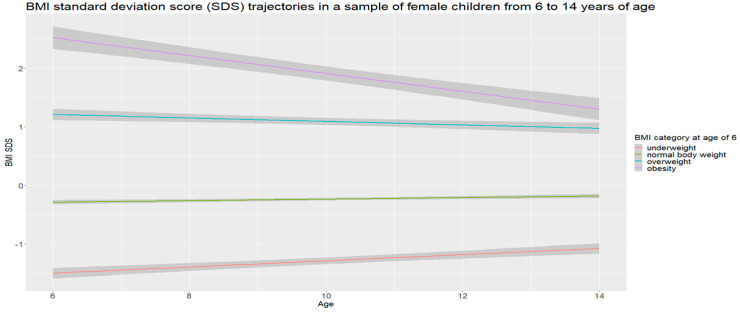
Girls.

**Figure 3 nutrients-16-04397-f003:**
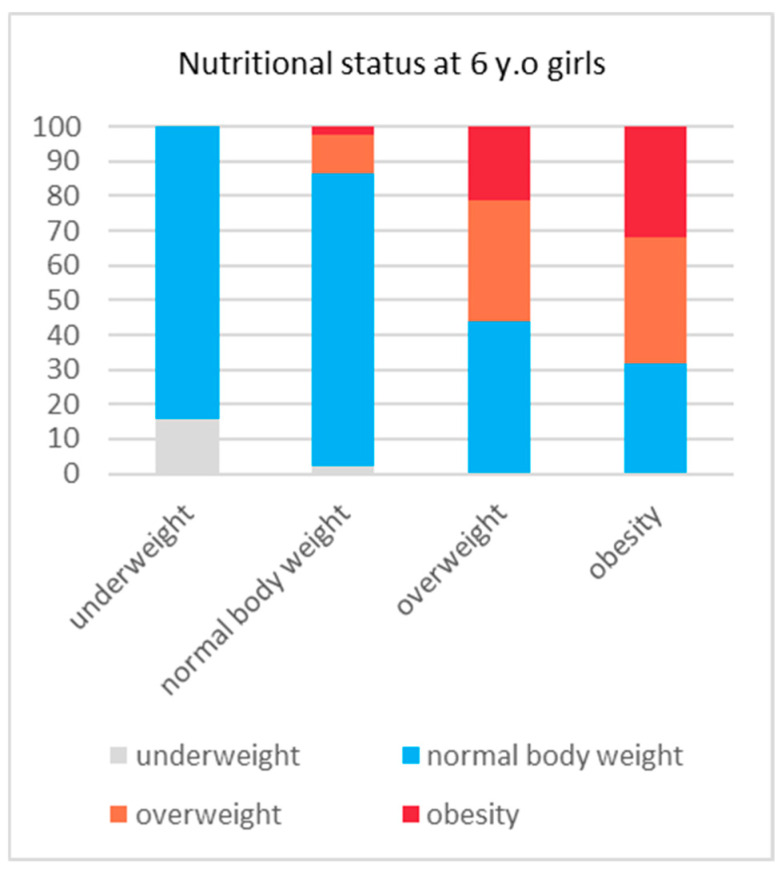
BMI at the 14 y.o. follow-up within BMI groups, as categorised at the age of 6 years in girls.

**Figure 4 nutrients-16-04397-f004:**
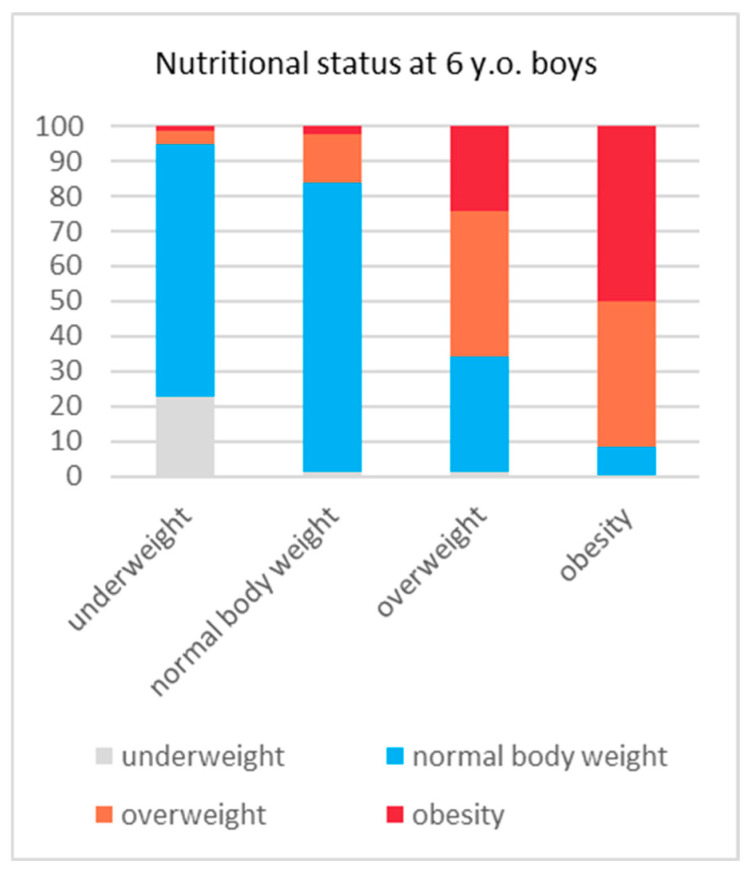
BMI at the 14 y.o. follow-up within BMI groups, as categorised at the age of 6 years in boys.

**Figure 5 nutrients-16-04397-f005:**
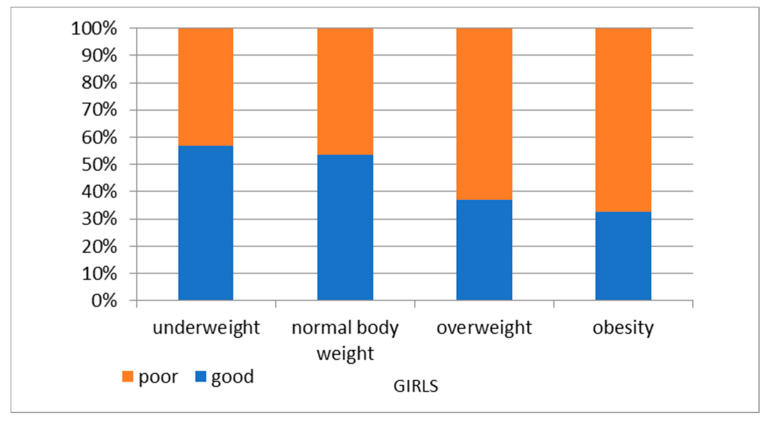
BMI categories of 14-year-olds by KPRT result at 6 years old—girls.

**Figure 6 nutrients-16-04397-f006:**
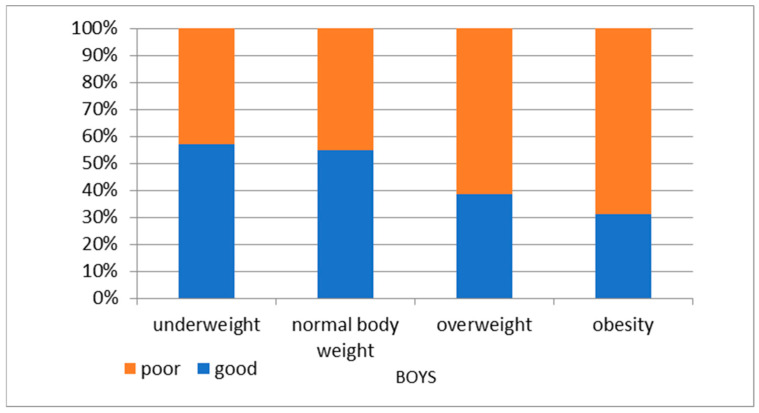
BMI categories of 14-year-olds by KPRT result at 6 years old—boys.

**Figure 7 nutrients-16-04397-f007:**
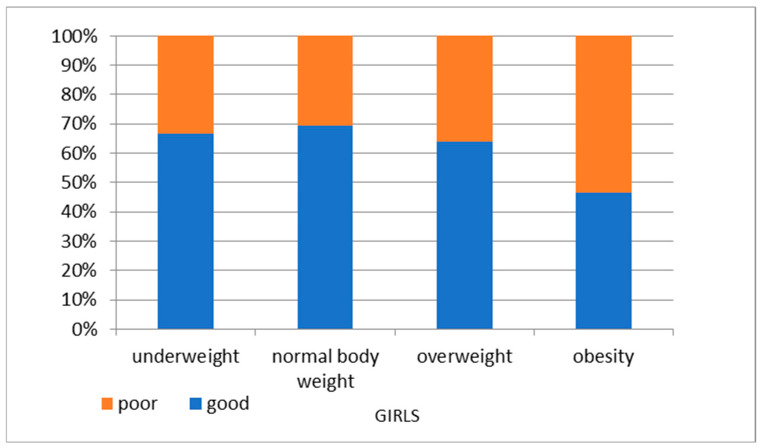
KPRT results at 14 years of age by BMI category at 6 years of age—girls.

**Figure 8 nutrients-16-04397-f008:**
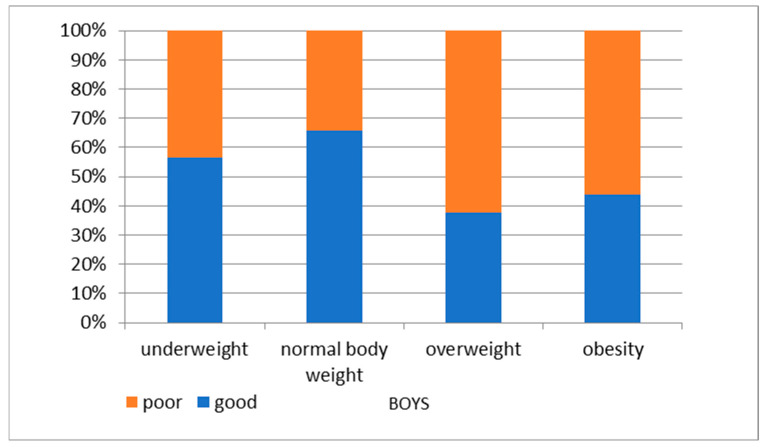
KPRT results at 14 years of age by BMI category at 6 years of age—boys.

**Table 1 nutrients-16-04397-t001:** BMI categories in study participants.

BMI Category	Underweight	Normal Body Weight	Overweight	Obesity	Total	*p*-Value
6 y.o., N (%) %girls/%boys	138 (4.49%) 45.65%/54.35%	2412 (78.44%) 49.17%/50.83%	336 (10.93%) 50.6%/49.40%	189 (6.15%) 48.15%/51.85%	3075 (100%) 49.11%/50.89%	*p* = 0.705
10 y.o., N (%) %girls/%boys	122 (3.97%) 54.10%/45.90%	2331 (75.80%) 49.46%/50.54%	427 (13.89%) 45.43%/54.56%	195 (6.34%) 49.74%/50.26%	3075 (100%) 49.11%/50.89%	*p* = 0.845
14 y.o., N (%) %girls/%boys	67 (2.18%) 52.24%/47.76%	2292 (74.54%) 50.61%/49.39%	505 (16.42%) 44.36%/55.64%	211 (6.86%) 43.13%/56.87%	3075 (100%) 49.11%/50.89%	*p* = 0.961

**Table 2 nutrients-16-04397-t002:** Longitudinal analysis of BMI groups categorised at the age of 6 years in girls.

BMI Category at 6 y.o.	Underweight (N = 63)	Normal Body Weight (N = 1186)	Overweight (N = 170)	Obesity (N = 91)	Total (N = 1510)
BMI at 10 y.o.					
Underweight, N (%)	27 (42.9%)	38 (3.2%)	1 (0.6%)	0 (0.0%)	66 (4.4%)
Normal body weight, N (%)	36 (57.1%)	1039 (87.6%)	68 (40.0%)	10 (11.0%)	1153 (76.4%)
Overweight, N (%)	0 (0.0%)	92 (7.8%)	69 (40.6%)	33 (36.3%)	194 (12.8%)
Obesity, N (%)	0 (0.0%)	17 (1.4%)	32 (18.8%)	48 (52.7%)	97 (6.4%)
BMI at 14 y.o.					
Underweight, N (%)	10 (15.9%)	25 (2.1%)	0 (0.0%)	0 (0.0%)	35 (2.3%)
Normal body weight, N (%)	53 (84.1%)	1003 (84.6%)	75 (44.1%)	29 (31.9%)	1160 (76.8%)
Overweight, N (%)	0 (0.0%)	132 (11.1%)	59 (34.7%)	33 (36.3%)	224 (14.8%)
Obesity, N (%)	0 (0.0%)	26 (2.2%)	36 (21.2%)	29 (31.9%)	91 (6.0%)

**Table 3 nutrients-16-04397-t003:** Longitudinal analysis of BMI groups categorised at the age of 6 years in boys.

BMI Category at 6 y.o.	Underweight (N = 75)	Normal Body Weight (N = 1226)	Overweight (N = 166)	Obesity (N = 98)	Total (N = 1565)
BMI at 10 y.o.					
Underweight, N (%)	27 (36.0%)	29 (2.4%)	0 (0.0%)	0 (0.0%)	56 (3.6%)
Normal body weight, N (%)	48 (64.0%)	1074 (87.6%)	51 (30.7%)	5 (5.1%)	1178 (75.3%)
Overweight, N (%)	0 (0.0%)	106 (8.6%)	80 (48.2%)	47 (48.0%)	233 (14.9%)
Obesity, N (%)	0 (0.0%)	17 (1.4%)	35 (21.1%)	46 (46.9%)	98 (6.3%)
BMI at 14 y.o.					
Underweight, N (%)	17 (22.7%)	13 (1.1%)	2 (1.2%)	0 (0.0%)	32 (2.0%)
Normal body weight, N (%)	54 (72.0%)	1015 (82.8%)	55 (33.1%)	8 (8.2%)	1132 (72.3%)
Overweight, N (%)	3 (4.0%)	168 (13.7%)	69 (41.6%)	41 (41.8%)	281 (18.0%)
Obesity, N (%)	1 (1.3%)	30 (2.4%)	40 (24.1%)	49 (50.0%)	120 (7.7%)

**Table 4 nutrients-16-04397-t004:** KPRT and blood pressure results and body weight in children at the ages of 6, 10, and 14 years old.

%/Age	6 y.o. Overweight	6 y.o. Obesity	10 y.o. Overweight	10 y.o. Obesity	14 y.o. Overweight	14 y.o. Obesity
RR	N = 325	N = 184	N = 427	N = 195	N = 504	N = 210
Normal %	56.9	47.8	42.6	30.8	50.2	66.7
Hypertension %	43.1	52.2	57.4	69.2	49.8	33.3
KPRT score	N = 248	N = 127	N = 303	N = 85	N = 346	N = 120
Good %	37.1	18.9	34.7	29.4	50.9	30.0
Poor %	62.9	81.1	65.3	70.6	49.1	70.0

**Table 5 nutrients-16-04397-t005:** Impact of KPRT results at 6 years of age on body weight in the future, in children obese at 6 years old.

KPRT at 6 y.o.	Good <= 3 (N = 24)	Poor 5 and 4 (N = 103)	Total (N = 127)
10 years/14 years			
Underweight	0 (0.00%)/0 (0.00%)	0 (0.00%)/0 (0.00%)	0 (0.00%)/0 (0.00%)
Normal body weight	2 (8.3%)/2 (8.3%)	10 (9.7%)/22 (21.4%)	12 (9.4%)/24 (18.9%)
Overweight	12 (50.0%)/14 (58.3%)	41 (39.8%)/39 (37.9%)	53 (41.7%)/53 (41.7%)
Obesity	10 (41.7%)/8 (33.3%)	52 (50.5%)/42 (40.8%)	62 (48.8%)/50 (39.4%)

## Data Availability

The original contributions presented in this study are included in the article/Appendix A. Further inquiries can be directed to the corresponding author.

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
