# Peer review of "Body Weight Changes During Childhood and Predictors of Excessive Body Weight in Adolescence—A Longitudinal Analysis"

_nutrients, 2024, doi:10.3390/nu16244397_

Round 1
Reviewer 1 Report
Comments and Suggestions for Authors
Interesting manuscript, which aims to: to analyse weight changes in children from preschool age to adolescence and to identify early predictors of excessive weight in adolescence, such as blood pressure and physical fitness observed in preschool children.
needs to be approved several changes:
1. in the results I think it is necessary to incorporate a table (table 1) with the sociodemographic characteristics of the schoolchildren.
2. to improve the size and resolution of figures 1 and 2.
3. How do figures 3 and 4 differ from tables 2 and 3?
4. Improve the format of the tables, it is not very neat, revise the information for the authors or compare with other papers published by the journal.
5. it would be a good idea to add a logistic regression, to associate being physically active as a protective factor against obesity in the different stages assessed.
4. Explain or hypothesize in the discussion what the results obtained mean.
Author Response
Dear Reviewer,
Thank you for your valuable comments. We have tried to answer all the questions and make corrections in the text.
- in the results I think it is necessary to incorporate a table (table 1) with the sociodemographic characteristics of the schoolchildren.
Thank you for your comment. The sociodemographic data we collected are already included in Table 1. All the children participating in the study were residents of a one city, Gdańsk. Due to the specific focus of our research, we did not aim to gather additional sociodemographic information. However, we believe the data provided are sufficient to contextualize our findings within the scope of our study.
- to improve the size and resolution of figures 1 and 2.
I have increased the size of Figures 1 and 2 and ensured their improved resolution. The updated figures have been included in the supplementary materials for your review.
- How do figures 3 and 4 differ from tables 2 and 3?
Thank you for your question. Tables 2 and 3 present detailed numerical data, including specific valuesand comparisons, which allow for a precise analysis of the results. In contrast, Figures 3 and 4 are designed to provide a visual representation of the trends and patterns derived from these data. The purpose of the figures is to facilitate a more intuitive understanding of the key findings, making it easier to identify relationships or differences at a glance. By including both detailed tables and illustrative figures, we aim to cater to readers who prefer different approaches to analyzing and interpreting the data.
- Improve the format of the tables, it is not very neat, revise the information for the authors or compare with other papers published by the journal.
Thank you for pointing this out. I have reformatted the tables in accordance with the journal's guidelines to ensure they are clearer and more visually organized. Please let me know if any further adjustments are needed.
- it would be a good idea to add a logistic regression, to associate being physically active as a protective factor against obesity in the different stages assessed.
Thank you for your interesting comments. We have created regression models and included them in the text and supplements.
Line 270-279
According to the accompanying supplements logistic regression models, it appears that a poor KPRT score significantly increases the risk of overweight and obesity in children-among 6-year-olds by more than 3 times ( odds ratio OR=3.2), 10 and 14-year-olds by more than 2.5 times (OR=2.58 10 y.o, OR=2.65 14 y.o), compared to children with a good KPRT score. The results indicate that low physical fitness is a key risk factor for overweight and obesity. It is worth paying attention to preventive measures that promote physical activity from an early age to improve NRPT scores and reduce the risk of weight problems. In addition, in the study group, being a boy is associated with a higher risk of EBW compared to girls, among 6-year-olds by 26%, 10 y.o by 17%, 14 y.o by 19%. May suggest the need to differentiate interventions by gender.
- Explain or hypothesize in the discussion what the results obtained mean.
Line 290-303
The long-term follow-up of the study cohort revealed that excessive body weight in early childhood tends to persist into adolescence, emphasizing the need for early intervention and health education. We found that obese children who were physically fit at age 6 were less likely to remain obese by age 10 and in adolescence. This suggests that KPRT assessment at age 6 could be a useful tool for identifying children in need of intervention programs. Early identification of at-risk children allows for timely and targeted strategies to prevent the long-term consequences of obesity.
These findings underline the importance of monitoring children's physical development from an early age to implement timely interventions and prevent obesity. Early intervention and health support are crucial in managing overweight and obesity in children. Promoting physical activity and healthy eating habits at a young age can help establish long-term healthy behaviors. Future research should focus on evaluating the effectiveness of early intervention programs to determine the most impactful strategies for combating childhood obesity.
We hope these additions address your concerns.
Once again, thank you for your helpful feedback.
Kind regards,
Aleksandra Lemanowicz-Kustra
Reviewer 2 Report
Comments and Suggestions for Authors
The manuscript ”Body weight changes during childhood and predictors of excessive body weight in adolescence – a longitudinal analysis” is well done and interesting.
The research aims to analyze weight changes in children from preschool age to adolescence and to identify early predictors of excessive weight in adolescence, such as blood pressure and physical fitness observed in preschool children. The manuscript is of value, the explanation of the factors influencing abnormal body weight is crucial in understanding the problem of childhood obesity.
The sample size is large, the analysis method is adequate, and the longitudinal analysis is of great value in this topic. The findings showing that children who were overweight or obese at age 6 had a higher risk of remaining so for a longer period are consistent with other findings from different countries.
The authors discuss their results based on well-selected literature findings and highlight the need for early intervention and health education because nutritional disorders are one of the key health, social, and psychological problems worldwide.
I have only one concern: Please include the limitations of the study.
Author Response
Dear Reviewer,
Thank you for your valuable suggestions.
Regarding your concern about including the limitations of the study, we have added the following points:
Line 300-309
The study was conducted in a single city, Gdansk, which may limit the generalizability of the results to children and adolescents from other regions or countries.
Although the study followed the children for several years (at ages 6, 10 and 14), a longer follow-up period might provide more insight into the long-term effects of early obesity and the effectiveness of interventions.
The study focused on physical fitness and blood pressure as predictors of overweight, but did not take into account other factors, such as socioeconomic status, diet or genetic predisposition, which might provide a more complete understanding of the causes of childhood obesity.
We hope these additions address your concerns.
Once again, thank you for your helpful feedback.
Kind regards,
Aleksandra Lemanowicz-Kustra